# Exploiting Cell-Based Assays to Accelerate Drug Development for G Protein-Coupled Receptors

**DOI:** 10.3390/ijms25105474

**Published:** 2024-05-17

**Authors:** Yuxin Wu, Niels Jensen, Moritz J. Rossner, Michael C. Wehr

**Affiliations:** 1Research Group Cell Signalling, Department of Psychiatry and Psychotherapy, LMU University Hospital, LMU Munich, Nussbaumstr. 7, 80336 Munich, Germany; 2Systasy Bioscience GmbH, Balanstr. 6, 81669 Munich, Germany; 3Section of Molecular Neurobiology, Department of Psychiatry and Psychotherapy, LMU University Hospital, LMU Munich, Nussbaumstr. 7, 80336 Munich, Germany

**Keywords:** GPCR, drug screening, drug discovery, cell-based assay, GPCR engineering

## Abstract

G protein-coupled receptors (GPCRs) are relevant targets for health and disease as they regulate various aspects of metabolism, proliferation, differentiation, and immune pathways. They are implicated in several disease areas, including cancer, diabetes, cardiovascular diseases, and mental disorders. It is worth noting that about a third of all marketed drugs target GPCRs, making them prime pharmacological targets for drug discovery. Numerous functional assays have been developed to assess GPCR activity and GPCR signaling in living cells. Here, we review the current literature of genetically encoded cell-based assays to measure GPCR activation and downstream signaling at different hierarchical levels of signaling, from the receptor to transcription, via transducers, effectors, and second messengers. Singleplex assay formats provide one data point per experimental condition. Typical examples are bioluminescence resonance energy transfer (BRET) assays and protease cleavage assays (e.g., Tango or split TEV). By contrast, multiplex assay formats allow for the parallel measurement of multiple receptors and pathways and typically use molecular barcodes as transcriptional reporters in barcoded assays. This enables the efficient identification of desired on-target and on-pathway effects as well as detrimental off-target and off-pathway effects. Multiplex assays are anticipated to accelerate drug discovery for GPCRs as they provide a comprehensive and broad identification of compound effects.

## 1. Introduction

G protein-coupled receptors (GPCRs) are seven transmembrane (7TM) cell surface receptors and major drug targets. GPCRs represent the largest protein family in humans, comprising 826 out of 20,283 (4%) genes [1]. In humans, the majority of GPCRs are olfactory receptors. The remaining ~350 non-olfactory members are considered druggable, of which 165 have been identified as drug targets [1]. Importantly, 35% of all marketed and FDA-approved drugs target GPCRs, making them the focus of intensive investment by pharmaceutical companies [2,3,4].

With the recent advances in multitarget drugs that regulate multiple GPCRs simultaneously [5] and the increasing need to screen for off-target activities [6], it is necessary to have efficient, robust, and reproducible assays to measure the activation of GPCRs in living cells. Furthermore, it is crucial to identify the signaling pathways that a specific GPCR/drug combination regulates to understand cellular physiology. In addition, it is increasingly important for drug discovery campaigns to assess both the desired on-target and on-pathway effects, as well as the potentially detrimental off-target and off-pathway effects. This review examines the currently available cell-based assays that evaluate GPCR activities and their downstream signaling pathways.

## 2. GPCR Structure and Function

GPCRs and GPCR-mediated signaling have been the focus of research since the pioneering work on adrenergic receptors by Robert J. Lefkowitz and Brian Kobilka, who won the 2012 Nobel Prize in Chemistry for their research on GPCRs [7]. A given GPCR can regulate multiple signaling pathways in different cell types, depending on the cellular transducer and effector repertoire. Therefore, GPCRs play a crucial role in signal transduction in diverse physiological responses, which include sensory perception, olfaction, neurotransmission and hormone signaling, immune response, cardiovascular regulation, and essentially every other physiological function. Likewise, GPCRs are implicated in the pathogenesis or pharmacological treatment of frequently occurring disorders, such as cancer [8,9], hypertension [10], diabetes [11], and mental disorders like schizophrenia and depression [12,13], supporting their key role in health and disease.

Heterotrimeric G proteins formed of Gα, Gβ, and Gγ subunits, together with GPCR kinases (GRKs) and β-arrestins, are the three main types of protein families that directly interact with GPCRs upon activation and are called transducers. Additionally, pathway selectivity in assays can be affected by the tested ligand, a concept called biased signaling or functional selectivity [14]. Agonist binding to a GPCR leads to conformational changes that transfer to the intracellularly bound heterotrimeric G protein and promotes a GDP/GTP exchange in Gα, leading to the dissociation of active Gα and Gβ/γ. These then initiate or modulate one of the classical second messenger cascades (cAMP, IP_3_/DAG, or cGMP) and can also directly interact with effector proteins such as ion channels, e.g., GIRK channels. GPCR activation also leads to receptor phosphorylation by GPCR kinases (GRKs), β-arrestin-mediated desensitization, and endosomal internalization (Figure 1A) [14]. Recent evidence suggests that this can lead to sustained intracellular G protein-mediated signaling from endosomes, resulting in MAP kinase (MAPK) pathway activation and nuclear gene regulation [15,16,17]. However, β-arrestins can also activate MAPK signaling directly at the plasma membrane [18,19].

GPCRs are often called ’7TM receptors’ because they consist of seven hydrophobic alpha-helical transmembrane domains (TM1–TM7) in a common three-dimensional arrangement. Each GPCR consists of a single polypeptide with an extracellular N-terminus, an intracellular C-terminus, three extracellular loops (ECL1–ECL3), and three intracellular loops (ICL1–ICL3). The orthosteric binding site for the endogenous ligand is located either in between the transmembrane helices, in the extracellular domain, or both [20]. Even an unusual intracellular binding site has recently been described for TAS2R14 [21]. When viewed from the extracellular side, the 7TM regions are arranged in an anti-clockwise orientation (Figure 1B) [20]. Agonist binding and GPCR activation result in a prominent outward movement of TM6 by 7–19 Å away from TM3, thereby breaking specific interactions (‘locks’) [20]. This conformational change is transferred via the directly connected ICL3 to the bound Gα protein and leads to the release of the GDP and the concomitant dissociation of the active G protein subunits followed by downstream signaling [22,23,24].

## 3. Biased Signaling Mechanisms of GPCRs

Biased signaling or functional selectivity describes the observation that different ligands can preferentially activate different pathways through the same receptor with regards to potency as well as efficacy [14]. Ligands can affect the coupling preferences of GPCRs for different G protein subtypes that modulate cAMP, calcium, or RhoA. Ligands can also affect GPCR phosphorylation via GRKs, arrestin binding, internalization, and, according to recent evidence, G protein-dependent endosomal signaling in a biased fashion. Indeed, agonists and antagonists that selectively promote either G protein-dependent or β-arrestin-dependent responses were identified in drug discovery campaigns [25,26]. This concept was further extended to different combinations of G protein subtypes [27]. However, the multiple alternative transducers and their various pathway components downstream of an activated GPCR contribute to signaling bias [14]. Biased signaling can become apparent at any hierarchical level of signaling, i.e., at the level of transducers, effectors, second messengers, and transcription factors (Figure 1C). Additionally, more recent reports suggest that proteins other than G proteins, GRKs, and β-arrestins can directly bind to active GPCRs and function as transducers, as discussed in more detail below [28,29,30,31]—another potential source of bias that might be exploited in the future.

Ligands can induce receptor conformations with biased preferences for GRKs, arrestins, and internalization. Ligands can then also have different endosomal retainment properties [32]. When using cell-based assays to monitor GPCR activation and downstream signaling, it is important to consider the potential complications arising from different binding and signaling kinetics under potentially non-equilibrium assay conditions [33,34].

### 3.1. Heterotrimeric G Proteins Are Initial Transducers of GPCR Signaling

The subunits Gα, Gβ, and Gγ of heterotrimeric G proteins dissociate into Gα and the Gβγ complex upon GPCR activation. Both Gα and Gβγ can initiate signaling through interactions with different downstream effector proteins (Figure 1D).

In humans, there are 16 Gα subunits that are grouped into four families with sequence homology and shared downstream signaling pathways: Gαs (Gαs and Gαolf), Gαi/o (Gαi1, Gαi2, Gαi3, Gαo, Gαz, Gαt1, Gαt2, and Gαgust), Gαq/11 (Gαq, Gα11, Gα14, and Gα15), and Gα12/13 (Gα12 and Gα13). While the Gαs family mediates the activation of adenylyl cyclase that synthesizes the second messenger cAMP, the Gαi/o family inhibits this enzyme. The Gαq/11 family mainly mediates the activation of phospholipase C, resulting in the second messengers IP_3_, DAG, and Ca^2+^. The Gα12/13 family is involved in the RhoA/Rho kinase signaling pathway. Signaling transduced by the Gα protein occurs not only in the plasma membrane but also in endosomes; e.g., beta-2-adrenergic receptor (ADRB2)–Gαs activation in endosomes stimulates cAMP synthesis and MAPK activation exclusively initiated from endosomes [16,17].

For Gβγ subunits, there are five types of β subunits and twelve different γ subtypes in humans [24,35]. These β and γ subunit subtypes can pair with each other to generate all the 60 theoretically possible unique βγ dimeric complexes. However, the actually possible number is limited by the expression profile of the cell and by the pairing preferences of certain subtypes [35,36].

Like for the selective coupling of a Gα subtype with a certain GPCR, specific Gβγ complexes preferentially associate with specific GPCRs [37]. Moreover, subtype-specific Gβγ complex-mediated signaling may be as prevalent as for Gα subunits, an aspect that has not yet received adequate attention [37]. In principle, each GPCR, once activated, liberates an Gα and Gβγ subunit per activation event. Similar to Gα, Gβγ complexes are initially anchored to the plasma membrane through the γ subunit and translocate to different organelles for signaling depending on the Gβγ subunits, such as the Golgi apparatus, early endosome, endoplasmic reticulum, or mitochondria [35]. However, various studies have suggested that Gβγ might be predominantly tied to Gi/o-coupled receptors, while the Gβγ of the other families may play a rather auxiliary role. The Gβγ subunits from Gi/o-coupled receptors were shown to regulate diverse effectors, including PLCβ2/3 [38], MAPKs [39], phosphoinositide-3-kinase (PI3K), AKT serine/threonine kinase, and voltage-gated calcium channels [38].

### 3.2. GPCR Kinases and Arrestins Mediate Biased Signaling

The human genome contains seven types of GRKs, which are grouped into three subfamilies: the GRK1 subfamily (GRK1 and GRK7), the GRK2 subfamily (GRK2 and GRK3), and the GRK4 subfamily (GRK4, GRK5 and GRK6) [24]. GRKs are serine/threonine kinases belonging to the AGC kinase superfamily [40]. Upon agonist activation, GPCRs recruit GRKs, which phosphorylate specific serine and threonine residues in intracellular domains of the receptor, particularly in ICL3 and the C-terminus [24]. Subsequently, arrestins can bind to the phosphorylated receptors.

There are four types of arrestin in the human genome, divided into two subfamilies: the visual subtypes (S-antigen visual arrestin (SAG) and arrestin 3 (ARR3)) and the non-visual subtypes (arrestin beta 1 (ARRB1) and arrestin beta 2 (ARRB2)) [41].

### 3.3. Non-Canonical Biased Signaling

In addition to signaling mediated by G proteins, GRKs, and arrestins, activated GPCRs can also bind to other proteins directly, such as 14-3-3 proteins. 14-3-3 proteins bind to phosphorylated GPCRs and act as adapters and regulators for a large number of proteins, including protein kinases, phosphatases, enzymes, and transcription factors [23]. 14-3-3 binding is dependent on GPCR phosphorylation, occurs after desensitization and internalization, and is slower than the GPCR/β-arrestin interaction. However, 14-3-3 signaling can be β-arrestin-independent, as shown for the β3-adrenergic receptor (ADRB3), which lacks a β-arrestin binding motif. In addition, different agonists can have varying potencies on 14-3-3 and β-arrestin signaling [29].

Cyclin-dependent kinase 5 (CDK5) physically interacts with the serotonin receptor 7 (HTR7A) and mediates G-protein-independent signaling [28]. Activated HTR7A, via CDK5, leads to hyperphosphorylation of the Tau mutant R406W and fibril formation, causing neural damage and death. In a corresponding mouse model of a tauopathy, this leads to reduced long-term potentiation (LTP) and impaired memory. By contrast, pharmacological inhibition of HTR7A abolished Tau phosphorylation, its aggregation, and the neurotoxic effects. Likewise, the genetic ablation of HTR7A restored LTP impairments and memory deficits. Therefore, this HTR7A/CDK5 signaling axis emerged as a new pathway for GPCR signaling involved in dementia and Alzheimer’s disease [28].

GPCRs can also interact directly with proteins of the cytoskeleton. For example, filamin A, an actin-binding protein that links actin filaments to membrane glycoproteins, interacts directly with several GPCRs, including the somatostatin receptor type 2 (SSTR2) [30,31]. Filamin A and SSTR2 interact in a transient manner, with filamin A restricting SSTR2 diffusion along actin fibers and promoting the agonist-dependent recruitment of SSTR2 into clathrin-coated pits and the subsequent internalization of SSTR2 [30]. The regulation of cytoskeleton proteins and their association with GPCRs can also be regulated by the formation and spatial arrangement of caveolae and signaling complexes [42]. For example, the pharmacological disruption of either actin filaments or microtubules has revealed that these elements restrict cAMP signaling by regulating the localization of GPCRs, G proteins, and adenylyl cyclases (ACs) [43].

## 4. Genetically Encoded Biosensors for Functional GPCR Assays

Genetically encoded assay techniques have been developed for measuring GPCR signaling in living cells at the level of the ligands, the receptors, the transducers, the effector and second messengers, and transcription (Figure 1C) [44].

### 4.1. Classification of GPCR Assays

GPCR assays are often categorized into real-time assays, often with fluorescent readouts, and reporter gene assays, which use transcriptional readouts and therefore have a delayed response of several hours [45]. GPCR assays are also classified by the interrogated cellular target. Lastly, assays can be divided into singleplex (separate) and multiplex (parallel) assays. In this review, we focus on cell-based GPCR assays that are genetically encoded and that can be used in singleplex and multiplex formats.

### 4.2. Singleplex GPCR Assays

Singleplex assays can only generate one data point from a single assay condition, while multiplex assays can generate several data points from one condition. The advantages of singleplex assays are often easier to design and establish, have a wider choice of detection methods, and are often faster. Several singleplex assays also allow for the real-time monitoring of receptor activity. Singleplex assays that are used to monitor GPCR activities and their downstream signaling are summarized in Table 1.

#### 4.2.1. Resonance Energy Transfer-Based Techniques

A subset of singleplex GPCR assays allows for the continuous real-time visualizations of GPCR activities. These usually apply genetically encoded conformational biosensors and biosensors based on Förster resonance energy transfer (FRET) and bioluminescence resonance energy transfer (BRET). This enables the dynamic monitoring of GPCR and G protein conformational states, subcellular location after GPCR activation, and the spatial kinetics of GPCR signaling at high temporal resolution (Figure 2) [47,53,69,70,71].

##### GPCR Activity Assays Using FRET and BRET

FRET biosensors rely on the non-radiative energy transfer of excitation energy from a UV-light excited-state donor-fluorophore to a nearby ground-state acceptor-fluorophore (Figure 2A) [72]. For FRET, the change in acceptor fluorescence is critically dependent on the close proximity of the donor and the acceptor, as the efficiency of FRET depends on the inverse sixth power of intermolecular separation [72]. The distance required for FRET to occur is less than 10 nm and therefore is well suited to examining direct molecular interactions. A FRET fluorophore pair needs to have (1) a sufficient separation of the excitation spectra, (2) an adequate overlap (>30%) between the emission spectrum of the donor fluorophore and the excitation spectrum of the acceptor fluorophore for energy transfer, and (3) a reasonable separation of the emissions for independent measurement of each fluorophore. A typical FRET pair is the cyan fluorescent protein (CFP), or one of its variants, as the donor and yellow fluorescent protein (YFP), or one of its variants, as the acceptor. FRET has been widely used to monitor dimerization of GPCRs and recruitment of direct binding partners such as G protein subunits and β-arrestins [49,52,55], as well as ligands [73]. For GPCR dimerization assays, the donor and the acceptor are fused to the two GPCRs. For recruitment assays, the donor is commonly fused to the GPCR, while the acceptor is coupled to the intracellular interaction partner.

BRET is a variant of FRET where the donor fluorophore is replaced by a luciferase. A common choice is *Renilla* luciferase (*R*luc) as the donor and YFP as the acceptor (Figure 2B) [74]. In comparison to FRET, BRET-based assays do not require an external UV light source to excite the donor, they have a low background, and they do not suffer from other limitations of FRET assays, such as photobleaching and autofluorescence.

For the original BRET1 assay system, coelenterazine was used as the substrate for *R*luc, which is converted into coelenteramide under emission of blue light with a peak at 475 nm. YFP, when in close proximity, is excited, resulting in a BRET emission of yellow light of 527 nm [75]. The intensity of light emissions by *R*luc and YFP are measured in a photometer, typically a plate reader, and the BRET signal is calculated as the ratio of GFP over *R*luc emissions. Several improvements to this first-generation BRET assay system have been made (Table 2). Initially, in BRET1, *R*luc/coelenterazine was used as the donor and YFP as the acceptor. The enhanced BRET2 (eBRET2) system uses the variant *R*luc8 and its substrate coelenterazine 400a as the donor and GFP2 as the acceptor, and currently represents the optimal BRET assay system. Compared to the wildtype *Renilla* luciferase, *R*luc8 has a four-fold improved light output due to eight favorable mutations [76]. As of today, BRET is one of the most applied assays to study the activation of GPCRs and their activation-dependent association with G proteins, GRKs, and β-arrestins [35,51,53,54,55,77].

BRET has also been used as a tool in GPCR Heteromer Investigation Technology (GPCR-HIT), which enables the identification of GPCR heteromer-specific pharmacology [78]. The BRET donor *R*luc is fused to the GPCR dimerization partner and the acceptor, such as YFP, can be a labeled ligand for competition assays or a transducer protein, such as β-arrestin-2, for measuring receptor activation.

Recently, BRET was used to establish a panel of biosensor assays, called TRUPATH, that measure the full complement of G protein-mediated signaling in living cells [54]. This, and an effector membrane translocation assay called EMTA [53], are the most comprehensive singleplex cell-based assays for GPCRs. TRUPATH uses *R*Luc8 fusions for all 14 human Gα subunits as donors and GFP2-tagged GPCRs as acceptors. Similarly, EMTA uses BRET2 to monitor the activation of twelve G protein subtypes based on the translocation of their synthetic effectors to the plasma membrane. EMTA also included additional β-arrestin BRET assays to profile the signaling bias of 100 therapeutically relevant human GPCRs.

**Table 2 ijms-25-05474-t002:** Features of BRET used in GPCR assays.

Assay	Donor	Acceptor	Substrate	Advantage	Disadvantage
BRET1 [74]	*R*luc	YFP	Coelenterazine	Strong signal, long lifetime	Poor spectral resolution due to small emission/excitation gap of only 45–55 nm
eBRET2 [53]	*R*luc8	GFP2	Coelenterazine 400a	5 to 30-fold increase in eBRET2 signal intensity and duration	None
BiFC-RET [79,80]	*R*luc	N-YFP and C-YFP	Coelenterazine h	Measures interaction between more than two proteins	The orientation of the test proteins fused to YFP fragments can prevent a proper complementation of YFP even when the test proteins are interacting
BiLC-RET [81]	*R*luc8-1 and *R*luc8-2 (complementation)	YFP/ mVenus	Coelenterazine h	Measures interaction between more than two proteins	The orientation of the test proteins fused to *R*luc8 fragments can prevent a proper complementation even when the test proteins are interacting

Abbreviations: BiFC, bimolecular fluorescence complementation; BiLC, bimolecular luciferase complementation; eBRET2, enhanced BRET2; GFP, green fluorescent protein; *R*luc, *Renilla* luciferase; YFP, yellow fluorescent protein; N-, N-terminal; C, C-terminal.

##### Conformational GPCR Assays Using FRET or BRET with GFP Superfolder Variants

Conformation-dependent GPCR-activation FRET sensors have been created by the fusion of a FRET donor to the intracellular loop ICL3 between TM5 and TM6 and a FRET acceptor to the C-terminus (or donor and acceptor switched) (Figure 2C) [82,83]. This approach has been used for various GPCRs, including α2A adrenergic receptor (ADRA2A) [84], β1 adrenergic receptor (ADRB1) [83], β2 adrenergic receptor (ADRB2) [85], parathyroid hormone 1 receptor (PTH1R) [82], and B2-bradykinin receptor (BDKRB2) [86]. Similarly, BRET-based biosensors were also developed using NanoLuc and HaloTag technology [77].

More recently, GPCR activation biosensors based on circularly permuted fluorescent proteins (cpFP) that replace ICL3 have been developed. The representative examples of this strategy are dLight [47] and GRAB-DA [48], which are fast biosensors for the dopamine receptors DRD1 to DRD5.

**Figure 2 ijms-25-05474-f002:**
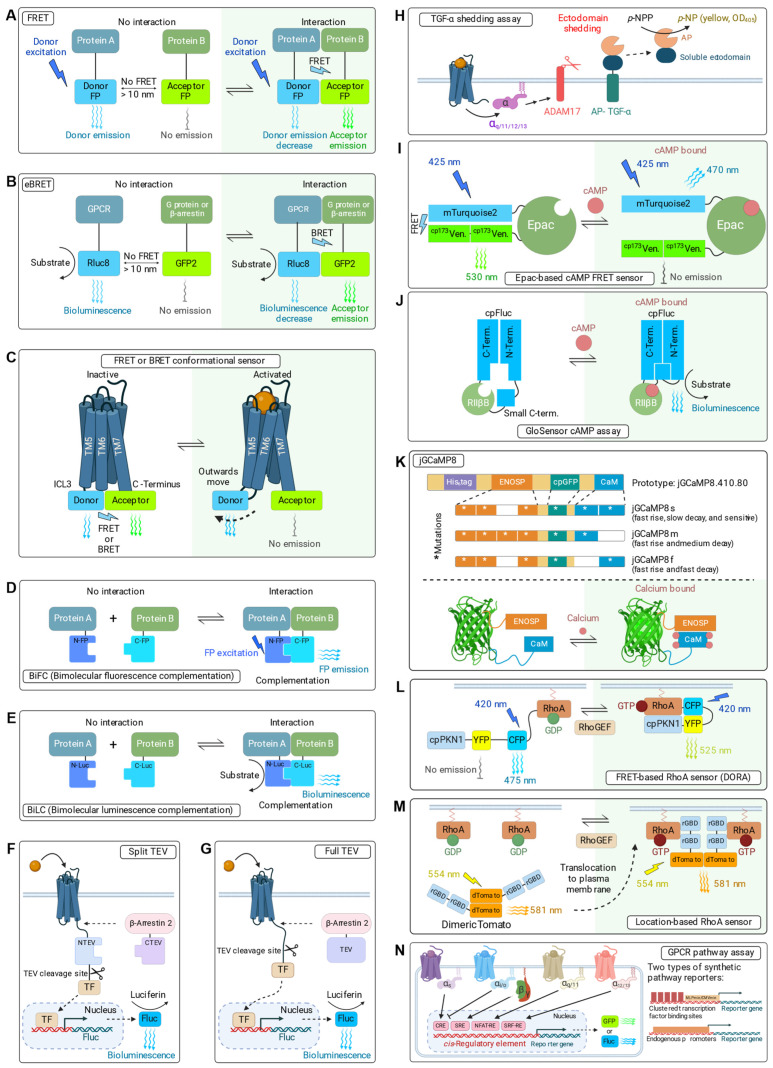
Genetically encoded methods to monitor GPCR activity and downstream signaling. (**A**) FRET assay for GPCR dimerization and G protein and β-arrestin recruitment. One candidate protein, e.g., the GPCR, is fused to CFP, while the other is fused to YFP. FRET only occurs when these proteins come into close contact. (**B**) BRET assay for GPCR dimerization and G protein and β-arrestin recruitment using enhanced BRET2 (eBRET2) with *R*luc8 as the BRET donor fused to the GPCR and GFP2 as the acceptor fused to the G protein or to β-arrestin. (**C**) Conformational FRET or BRET sensors where donor and acceptor are separated upon GPCR activation and the outwards movement of TM6 [23,77]. (**D**) Bimolecular fluorescence complementation (BiFC) assay. A split fluorescent protein, consisting of the two fragments N-FP and C-FP that are fused to the interacting proteins, e.g., GPCR and β-arrestin, folds into a functional fluorescent protein when the proteins come into close contact. (**E**) Bimolecular luminescence complementation (BiLC) assay. Same principle as in (**D**) but using a luciferase instead of a fluorescent protein. (**F**) Split TEV GPCR/β-arrestin-2 recruitment assay. A GPCR is fused to the N-terminal fragment of the TEV protease (NTEV), a TEV protease cleavage site, and an artificial transcription factor (TF), such as GAL4-VP16. The C-terminal fragment of the TEV protease is fused to β-arrestin-2. Upon GPCR activation, β-arrestin-2 is recruited, and NTEV and CTEV fragments fold into an active protease that cleaves the TF. The released TF translocates to the nucleus and binds to its recognition sequences to activate the transcription of a reporter gene, such as firefly luciferase [58]. (**G**) Full TEV β-arrestin-2 recruitment assay (Tango). Similar to the split TEV β-arrestin-2 recruitment assay in (**F**) but with a full-length TEV protease fused to β-arrestin-2. In full-TEV assays, the synthetic tTA transactivator protein is usually used as TF [57]. (**H**) TGF-α shedding assay to monitor GPCR/G protein coupling. Gαq/11/12/13 activates the metallopeptidase ADAM17 that cuts the TGF-α ectodomain fused to alkaline phosphatase (AP). Liberated AP can now convert colorless *para*-nitrophenylphosphate (*p*-NPP) to yellow *para*-nitrophenol (*p*-NP) for quantification [59]. (**I**) Fourth-generation Epac-based FRET sensor for monitoring cAMP. Binding of cAMP changes the conformation of the cAMP-binding protein Epac and brings mTurquoise2, a CFP variant with increased brightness and photostability, and a tandem of cp173Venus, a circular permutated version of the fluorescent protein Venus, into close contact with FRET [87]. (**J**) GloSensor assay to monitor cAMP. This sensor is based on a permutated split firefly luciferase where the two subunits are separated by the cAMP-binding domain RIIβB of protein kinase A (PKA). The RIIβB domain is located near the hinge region of the luciferase. The binding of cAMP to the RIIβB domain induces a conformational shift and leads to a functional luciferase [64]. (**K**) GCaMP calcium sensors. The depicted set of specialized calcium sensors is based on the genetically encoded calcium indicator jGCaMP8, which consists of the endothelial nitric oxide synthase (eNOS) calmodulin (CaM)-binding peptide (ENOSP), a circularly permutated GFP (cp-GFP), and of CaM. The binding of calcium to CaM results in the association with ENOSP and the functional activation of cpGFP. Specific mutations produce sensors that are optimized for a fast response with different decay times and sensitivities [66]. (**L**) FRET-based Dimerization-Optimized Reporter for Activation (DORA) RhoA sensor. This sensor for Gα12/13 activation is based on a circularly permuted RhoA interactor domain from protein kinase C-related kinase 1 (cpPKN1). When RhoA is activated by RhoGEF, cpPKN1 binds to the active RhoA, resulting in increased FRET efficiency [88]. (**M**) Location-based RhoA sensor with dimericTomato-2xrGBD domains. Another sensor for Gα12/13-dependent RhoA activation has been created by fusing dimeric tomato (dTomato), a variant of DsRed fluorescent protein. Mouse rhotekin G protein-binding domain (rGBD) binds active RhoA, resulting in fluorescence enrichment at the plasma membrane. The fluorescence signal is further enriched by the dimerization of dTomato, resulting in a complex containing two dTomato and four rGBDs. When RhoA is activated by Rho guanine nucleotide exchange factor (RhoGEF), the dimericTomato-2xrGBD will relocate to the active RhoA at the plasma membrane [68]. (**N**) GPCR pathway assays. Pathway activities following GPCR activation can be monitored by endogenous transcription factors that bind to synthetic reporters and initiate the expression of reporter genes, e.g., a luciferase. Synthetic reporters either consist of transcription factor binding sites clustered before a minimal promoter, such as the minimal adenoviral promoter (MLPmin) or the minimal CMV promoter (CMVmin), or they consist of endogenous promoter sequence fragments.

#### 4.2.2. Split Fluorescent Protein Assays

Split fluorescence protein assays, also known as bimolecular fluorescence complementation (BiFC) assays, are commonly used to assess the oligomerization of GPCRs, their interaction with G protein subunits, and with β-arrestins [89]. In BiFC assays, the fluorescent protein (FP) is split into two halves that only fold into a functional FP and produce a fluorescent signal when in close proximity (Figure 2D). Specifically, for the first BiFC assay, the GFP molecule was split into an N-terminal fragment (1–157) and a C-terminal fragment (158–238) [90]. The fluorescent protein Venus is a derivative of YFP that is optimized for biological cell applications [91] and has been frequently used to study GPCR dimerization in living cells, e.g., for the dopamine receptor DRD2 with the adenosine A2a receptor (ADORA2A) [79], for DRD2 with the cannabinoid receptor 1 (CB1) [80], and for the neuropeptide Y1 and Y5 receptors (NPY1/NPY5) [92]. BiFC using split Venus was also recently used to study all theoretically possible 60 non-optical Gβγ subunit complexes [35]. Several slightly differing split Venus variants were used, including 1–152 for the N-terminal fragment (VN) and 156–240 for the C-terminal fragment (VC) [93], 1–158 (VN) and 159–239 (VC) [94], 1–172 (VN) and 155–238 (VC) [79,80], and 1–172 (VN) and 156–239 (VC) [95]. The applications of split Venus for BiFC in GPCR assays have been excellently reviewed elsewhere [89].

The engineering of superfolder variants for GFP resulted in the identification of split GFP fragments that consist of the β-strands 1–10 and 11 (GFP1–10/GFP11 system) to form a bipartite split GFP system and of β-strands 1–9, 10, and 11 (GFP1-9/GFP10/GFP11) to form a tripartite system [96,97]. In the tripartite system, the β10 and β11 fragments were fused to the interacting proteins, while β1-9 was co-expressed as a detector. The tripartite system has the advantage of absent autofluorescence, presumably due to the reduced self-assembly of β10 and β11. In a fluorogenic assay using the tripartite superfolder GFP system, the activation of various GPCRs, including PAR1, PAR2, and ADRB2, was assessed using a β-arrestin recruitment assay in HEK293 cells [98]. In this assay, β11 was fused to the C-terminus of the GPCR, while β10 was linked to the N-terminus of β-arrestin. When these two fusion proteins were co-expressed with β1–9, the addition of an agonist led to increased green fluorescence. A red fluorescent protein, such as mCherry or mRuby2, can be co-transfected as a control for transfection efficiency.

#### 4.2.3. Split Luciferase Assays

Like FPs, luciferases, such as firefly, *Renilla*, and NanoLuciferase, can also be split into two fragments to assess protein/protein interactions. These assays are known as bimolecular luciferase complementation (BiLC) assays. In a split luciferase assay, the N-terminal and C-terminal fragments of the luciferases are fused to the test proteins. Once these interact, the luciferase fragments come into close proximity and fold into a functionally active complemented luciferase that converts its substrate to produce light, which can be quantified in a luminometer (Figure 2E). For GPCR interactions, split luciferase reporter approaches have not been used to the same extent as split fluorescent proteins, despite being equally able to rapidly detect protein/protein interactions. Several pharmacological studies on the chemokine receptors CXCR4 and CXCR7 used the split firefly luciferase system [99,100,101]. The split *Renilla* luciferase assay was used to assess the homodimerization of CXCR4 [102]. An improved form of *Renilla* luciferase, *R*luc8, was used to study DRD2 oligomerization [93]. NanoBit (Nanoluciferase Binary Technology), the most extensively used luminescent protein complementation assay, was developed by Promega [103]. The NanoLuciferase is a small reporter of 19 kDa in size and has been split into a larger fragment of 18 kDa and a small fragment of 1.3 kDa. Both fragments have a very low affinity for each other, minimizing background activity. NanoBiT has been used for the recruitment of β-arrestins [104,105,106] and G proteins to GPCRs [107].

#### 4.2.4. Split TEV GPCR β-Arrestin-2 Recruitment Assays

The split TEV GPCR β-arrestin-2 recruitment assay relies on the functional complementation of TEV protease fragments (Figure 2F) [58,108]. It monitors the activation of a GPCR at the membrane through β-arrestin-2 recruitment. The assay was designed as a singleplex assay but can also be adapted for use in multiplex assays. The tobacco etch virus (TEV) protease recognizes a specific cleavage site and can be split into N- and C-terminal fragments, called NTEV and CTEV. The NTEV fragment is fused to the C-terminus of GPCRs, while the CTEV fragment is linked to the C-terminus of a truncated form of ARRB2 (β-arrestin-2). Upon GPCR activation, the recruitment of ARRB2 brings NTEV and CTEV into close proximity, facilitating functional complementation of the TEV fragments into an active enzyme with proteolytic activity, which cuts into a TEV cleavage site to release the synthetic co-transcriptional activator GAL4-VP16 (GV) to initiate the transcription of a reporter gene, such as firefly luciferase [58]. Split TEV GPCR assays were further optimized for transient transfection assays for use in HEK293, PC12, HeLa, and U-2 OS cells. These assays demonstrated that, with the exception of the DRD2 receptor, an N-terminal signal peptide (SP) and a vasopressin 2 receptor tail (V2R tail) at the C-terminus of the construct, common in similar assay systems, can be omitted in split TEV GPCR assays [108].

#### 4.2.5. Full TEV GPCR β-Arrestin-2 Recruitment Assays

Like split TEV assays, the full TEV GPCR assay, also known as Tango assay, also uses a β-arrestin-2 recruitment strategy and is suited for multiplex assays [57]. In full TEV GPCR β-arrestin-2 recruitment assays, the GPCR is linked via a TEV cleavage site to the synthetic co-transcriptional co-activator tTA (GPCR-tcs-tTA), while the full-length TEV is fused to the C-terminus of a truncated β-arrestin-2 (Figure 2G). Once an activated GPCR recruits a β-arrestin-2, the TEV protease cuts the tTA off, resulting in the transcriptional activation of a firefly reporter gene. This technique has been applied to more than 300 GPCRs (PRESTO-Tango), and all plasmids encoding the GPCR-tcs-tTA open reading frames are available from Addgene (https://www.addgene.org/kits/roth-gpcr-presto-tango/ (accessed on 12 May 2024)), thus providing a valuable resource for academic research [48]. A variation of this assay uses a permutated firefly luciferase with an internal TEV protease cleavage site [29]. The TEV protease and the permutated firefly luciferase can both be fused to the GPCR or the interacting protein. When these come into close proximity, the TEV protease cleaves the permutated firefly luciferase, enabling the high-affinity luciferase fragments to self-complement into an active enzyme.

#### 4.2.6. TGF-α Shedding Assay

Another GPCR assay is the transforming growth factor-α (TGF-α) shedding assay (Figure 2H) [59]. The coupling profiles of 148 GPCRs to the Gαq wild-type and ten chimeric G proteins were measured. This enabled the identification of GPCR sequence-encoded features underlying G protein selectivity. In addition, the authors presented the first two Gα12-specific DREADDs (designer receptors exclusively activated by designer drugs), engineered receptors that respond to a synthetic ligand. Previously, DREADDs were only available as the Gαs, Gαi/o, and Gαq/11-coupled M3Ds, M4Di, and M3Dq GPCRs [109]. Therefore, this TGF-α shedding assay has extended the availability of DREADDs to all four Gα protein subfamilies.

#### 4.2.7. Assays for Second Messengers and Effectors

Downstream signaling dynamics of second messengers activated by GPCRs can be monitored by FRET. cAMP is a second messenger that is raised by Gαs and decreased by Gαi. cAMP levels can be measured through genetically encoded biosensors based on the protein Epac (exchange protein directly activated by cAMP) [61]. Epac1 and Epac2 contain a cAMP binding site and are guanine nucleotide exchange factors (GEFs) that activate signaling of the small GTPases Rap, Rap1, and Rap2 [110]. The conformational change induced by cAMP binding to Epac has been utilized to create FRET biosensors. Epac-based cAMP biosensors of the fourth generation have an optimized signal-to-noise ratio, outstanding photostability, and a high dynamic range. They use mTurquise2 as the FRET donor and two molecules of circularly permutated ^cp173^Venus as the FRET acceptor (Figure 2I) [87]. Both ‘gain-of-FRET’ variants of Epac biosensors for studying Gαs-coupled GPCRs (resulting in increased cAMP levels) and ‘loss-of-FRET’ variants for Gαi-coupled GPCRs (resulting in decreased cAMP levels) exist [62].

cAMP levels can also be measured with the commercial GloSensor assay from Promega (Figure 2J). This biosensor uses a *Photinus pyralis* firefly luciferase as the readout [111]. The GloSensor has a fast response to changes in cellular cAMP levels and can be used for real-time monitoring [112]. Structurally, the GloSensor is based on a permutated firefly luciferase into which the cAMP-binding domain of the human type II-beta regulatory subunit of PKA (RIIβB) has been inserted near the hinge region of the luciferase [64]. The binding of cAMP to the RIIβB domain leads to a conformational shift, resulting in the functional complementation of the luciferase domains. The active enzyme then converts its substrate luciferin to oxyluciferin under emission of a yellow-green light with a peak at 562 nm. The GloSensor assay offers several advantages over genetically encoded FRET and BRET biosensors, including high sensitivity, a broad dynamic range [113], and fast kinetics [112,114]. Because of this, this assay is often used for high-throughput screening at the singleplex level.

Calcium signaling and calcium flux are initiated by GPCRs linked to Gαq, such as the HTR2A and the muscarinic M3 receptors. Genetically encoded calcium indicators (GECIs) can be used to monitor calcium-dependent fluorescence in living cells [115]. Particularly, GFP-based GCaMP sensors have been iteratively engineered for an enhanced signal-to-noise ratio, which makes them suited for the real-time measurement of fast calcium dynamics, such as calcium flux in living cells [116]. Modern GCaMPs consist of a calcium-binding domain, such as the M13 domain, linked to a circularly permuted fluorescent GFP. Over the years, various generations of GCaMPs have been developed, each with improved characteristics, such as increased brightness, faster kinetics, and enhanced calcium sensitivity. For instance, GCaMP6 variants have become widely used due to their superior performance in detecting small calcium changes in neurons and other cell types [65]. Particularly, the GCaMP6f (fast) variant was optimized for kinetic measurements of action potential in neurons [65]. Recently, jGCaMP8 was reported as a further optimized GCaMP with ultra-fast kinetics (half-rise times of 2 ms) (Figure 2K) [66]. The use of genetically encoded calcium sensors for screening assays that measure calcium flux is less common. One example is the use of the GCaMP3 variant dCys-GCaMP for measuring the Gαq-coupled adrenergic receptor ADRA1A [117]. However, with the development of optimized GCaMPs with improved signal-to-noise ratio and ultra-fast kinetics, like jGCaMP8, the use of GCaMPs for GPCR assays is expected to rise [118].

As an alternative to GFP-based GCaMPs, the red fluorescent R-GECOs were designed as genetically encoded calcium indicators, which allow for duplex assays together with an independent green or yellow sensor, e.g., for cAMP [119]. R-GECO has been used for measuring calcium signaling in HeLa cells expressing the dopamine receptors DRD1 to DRD5 and the serotonin receptors HTR1 to HTR7 [120]. Especially, the Gαq-coupled receptors HTR2A and HTR2C displayed a strong response upon serotonin stimulation.

GPCRs coupled to Gα12/13, such as sphingosine-1-phosphate and thrombin receptors, signal through RhoA [121]. RhoA is a GTPase that regulates cytoskeleton remodeling, cell adhesion, cell migration, and proliferation. RhoA switches between an inactive GDP-bound state and an active GTP-bound state. It is activated by a GDP-to-GTP exchange that is facilitated by Rho guanine exchange factors (RhoGEFs) [122]. RhoA activity can be monitored either by FRET-based biosensors (Figure 2L) or location-based biosensors (Figure 2M) [123]. Both types of RhoA biosensors contain a small G protein-binding domain (GBD), which binds to the active GTP-bound state of RhoA. A FRET-based RhoA biosensor uses a circularly permuted RhoA interactor domain from protein kinase C-related kinase 1 (cpPKN1), which binds to active RhoA [88]. When N-terminal RhoA is activated by RhoGEF, the C-terminal cpPKN1 binds to active RhoA, thereby forcing the FRET pair into close proximity, resulting in increased FRET efficiency. While FRET-based sensors require higher excitation energies with a risk of photobleaching and phototoxicity, the location-based sensors are of a simpler design and more precisely indicate the subcellular location of RhoA activity, but they suffer from a higher background activity. However, a recently developed RhoA biosensor that uses a dimerized rothekin GBD (rGBD) fused to dTomato (dimericTomato-2xrGBD) shows an improved signal-to-noise ratio and higher sensitivity when measured in HeLa cells [68]. Mechanistically, this was explained by a high avidity of the interaction of the sensor with RhoA, facilitated by the duplicated rGBD RhoA-binding domain and the dimerization of dTomato.

#### 4.2.8. GPCR Pathway Assays

The activity of GPCRs can also be captured at the transcriptional level by reporter gene assays (Figure 2N). In general, reporter gene assays are very sensitive and robust and can be easily applied to high-throughput screening approaches. These reporters commonly consist of clustered transcription factor binding sites that are linked to a minimal promoter, such as the adenoviral major late promotor or the minimal cytomegalovirus (CMV) promotor, to drive a reporter gene such as GFP or luciferase. Single transcription factors that bind to their transcription factor binding sites are used as surrogate markers of upstream pathway activation and represent the endpoints of signaling cascades that are initiated at the GPCR. Such synthetic reporters have been designed for many pathways, including those downstream of the four subgroups of Gα proteins (Figure 1D) [45]. Gαs stimulates adenylate cyclase (AC), resulting in an increase of cAMP and the downstream activation of the cAMP response element (CRE) reporter through the cAMP/CREB pathway. In contrast, Gαi inhibits AC, leading to a reduction of cAMP and the concomitant decrease of CRE reporter activity. Gαq/11 stimulates phospholipase C (PLC) leading to the release of calcium from intracellular stores and activation of the transcription factor Nuclear Factor of Activated T cells (NFAT) that binds to the NFAT response element (NFAT-RE). In addition, calcium also activates the CRE reporter through CREB. Gβγ, released from Gαi, activates the ERK branch of the MAPK pathway, resulting in the activation of the serum response element (SRE). Gα12/13-coupled receptors signal through RhoA and the serum response factor (SRF) to activate the SRF response element (SRF-RE).

As an alternative to clustered transcription factor binding sites, endogenous promoter sequence fragments, which contain multiple transcription factor binding sites, are used in synthetic pathway sensors to drive a reporter gene. For example, the promoter of the early growth response 1 (EGR1p) contains five SRE sites, one AP1 site, two SP1 sites, two CRE sites, and one EGR1 site (driving its own repression) [124,125] and can be used to sensitively monitor activities in the ERK branch of MAPK signaling [126].

### 4.3. Multiplex GPCR Assays

Multiplex assays can simultaneously generate multiple independent readouts from the same sample. As for singleplex assays, multiplex assays can monitor GPCR activities at different levels of the intracellular signaling cascades. Simple multiplexing techniques use mutually compatible fluorescent or luminescent readouts that have no spectral overlap. However, the multiplexing complexity can be significantly higher when using transcriptional barcode reporters (Table 3).

Barcodes are short stretches of nucleotides (DNA or RNA) and therefore can have an essentially unlimited sequence diversity [127]. Next-generation sequencing (NGS) can simultaneously detect the full repertoire of short barcodes from a single sample, as well as from many pooled samples simultaneously [128]. The combination of molecular barcodes with the still-increasing availability of NGS, even in smaller laboratories, contributes to the recent success of barcoded multiplex assays.

Based on the position in the signaling cascade that is targeted, barcoded GPCR assays can be further divided into two types, such as barcoded GPCR receptor assays (Figure 3A) and barcoded GPCR pathway assays (Figure 3B). However, as both types of barcoded GPCR assays use genetically encoded barcodes as transcriptional readouts, both types can easily be combined.

**Table 3 ijms-25-05474-t003:** Genetically encoded multiplex assays for monitoring GPCR activity and signaling.

Signaling Level	Target Mechanism	Assay Technique	Readout	References
Receptor	Dimerization	BiFC	Fluorescence	[79,80]
Receptor	Dimerization	BiFC, eBRET2	Fluorescence, Luminescence	[129]
Receptor + Transducer	Dimerization + β-arrestin recruitment	BiLC, eBRET2	Fluorescence, Luminescence	[81]
Transducer	β-arrestin-2 recruitment	Split TEV assay	NGS (barcodes)	[130]
	β-arrestin-2 recruitment	Full TEV assay	NGS (barcodes)	[131]
Transcription	Transcription factors	Pathway assay	NGS (barcodes)	[132]
Transducer + transcription	β-arrestin-2 recruitment + transcription factors	Split TEV + Pathway assay	NGS (barcodes)	[133]

Abbreviations: BiFC, bimolecular fluorescence complementation; BiLC, bimolecular luciferase complementation; eBRET2, enhanced BRET2; TEV, tobacco etch virus protease.

#### 4.3.1. Barcoded GPCR Receptor Assays

Barcoded GPCR receptor assays monitor the activity of GPCRs through the recruitment of directly interacting proteins to activated receptors [80,92]. The first barcoded GPCR receptor assay that was published in 2018 is the GPCRprofiler. This split TEV GPCR β-arrestin-2 recruitment assay was conducted in transiently transfected U2OS and PC12 cells for monitoring the activity of 19 different GPCRs (Table 3) (Figure 3A) [130]. This assay confirmed the cross-activation of GPCR subfamilies by aminergic ligands such as dopamine and epinephrine and identified new targets of the neuroleptics paliperidone and aripiprazole. Transient transfection assays can significantly reduce assay development times and provide high flexibility and are therefore a good option for proof-of-concept assays or for testing novel targets. However, using transient transfections may lead to a higher variability due to changes in transfection efficiency, the over-expression of test proteins, an increased background, and handling effects, which can all reduce assay robustness [93,94,95,96]. To eliminate, or at least minimize these limitations, some or all components of split TEV assays, e.g., the β-arrestin-2-CTEV chimera and/or the GPCRs linked to the NTEV fragment, can be stably integrated into assay cells. Similarly, the PRESTO-Tango platform, which uses the full TEV protease in a β-arrestin-2 GPCR recruitment assay, was developed into a multiplex GPCR screening platform called PRESTO-Salsa. A total of 314 GPCRs were initially screened in a transient co-transfection assay for modulators in a 96-well format using an NGS readout [131]. As purely transient assays gave a high background, a stable HEK293T cell line expressing the chimeric β-arrestin-2-TEV protein was generated and used to screen 314 GPCRs in barcoded reporter gene assays in a 96-well format, where GPCRs and barcode reporter plasmids were transiently transfected into the stable screening cell line. In addition, the NGS readout gave a better signal-to-noise ratio than EGFP reporter fluorescence.

#### 4.3.2. Barcoded GPCR Pathway Assays

GPCR activation can be monitored by reporter gene assays using *cis*-regulatory elements that capture the activities of transcription factors and drive molecular barcode expression. These transcription factors can be considered endpoints of cellular signaling cascades and capture the activities of pathways of GPCR activation (Figure 3B) [45,71]. Several dimensions of multiplexing are possible by combining assay, well, pathway, receptor, and cell-specific barcodes [127]. Due to the signal amplification effects of the complex signaling cascades, reporter gene assays offer a high signal-to-background ratio and low variability [71]. Using this method, various cellular pathways downstream of HTR2A activation, such as CRE, NFAT-RE, and MAP kinase-regulated gene transcription, were measured simultaneously in a transient barcoded assay [133]. Similarly, 39 orphan Gαs-coupled odorant GPCRs that increased cAMP and stimulated a CRE sensor were tested simultaneously against 181 odorants in a barcoded pathway assay, thereby identifying 79 new interactions [132]. In this assay, each GPCR was stably integrated into HEK293 cells, along with a CRE sensor linked to a specific molecular barcode. As each GPCR was linked to a unique barcode, responses to each test compound were analyzed simultaneously from 39 cell lines pooled into a single well.

#### 4.3.3. Multiplex BiFC and BiLC Assays

Multiplex BiFC assays, also known as multicolor BiFC assays, have been used to study the interaction of one GPCR with two others simultaneously. Split Cerulean, a variant of CFP, was combined with split Venus. GPCR A was tagged with the C-terminal part of Cerulean (CC), while GPCR B was tagged with the N-terminal part of Venus (VN), and GPCR C was fused with the N-terminal fragment of Cerulean (CN). CC can complement VN and CN to produce yellow Venus (VN/CC) and turquoise Cerulean (CN/CC) fluorescence simultaneously within the same cell. This was used to study the hetero- and homodimerizations between ADORA2A and DRD2 [79], as well as between CB1 and DRD2 [80].

BiFC assays can also be combined with BRET assays to study the formation of trivalent and tetravalent signaling complexes. For example, *R*luc8 was fused to AC as a donor, while the Venus subunits were fused to Gβ and Gγ, acting together as the acceptor when complemented after ADRB2 receptor activation, enabling the measurement of a GPCR signaling complex [134]. Similarly, a trivalent complex can be measured using a BiLC assay with split *R*luc8 as the donor and Venus as the acceptor. This was shown for DRD2 and ADORA2A fused to split *R*luc8 fragments, while β-arrestin was fused to YFP [81]. In a continuation of this principle, BiLC and BiFC were combined to assess the existence of tetravalent interactions [134].

Multicolor BiFC assays and BiLC assays can also be used for live-cell imaging. However, the multiplexing capacity is limited to the number of distinguishable and independent fluorescent proteins and luciferases. Up to six different luciferases could be expressed from one plasmid in transient transfection assays and measured simultaneously [135]. This multiplex sextuple luciferase reporter contained five pathway reporters, including a MAPK-signaling reporter and a control reporter. Although not yet applied to the GPCR signaling context, the multiplex luciferase reporter could easily be adapted to include CRE and NFAT-RE reporters to monitor GPCR signaling.

## 5. Advantages and Limitations of Genetically Encoded Reporter Systems

The overexpression of any signaling component, such as GPCRs, G proteins, or beta arrestins in genetically encoded reporter systems, may regularly result in alterations of cellular signaling [136]. For instance, a BRET assay using a specific GPCR and Gαs protein may artificially elevate cAMP levels via the activation of AC, resulting in augmented downstream signaling and transcriptional responses. Nevertheless, genetically engineered assays using the overexpression of proteins are an effective means of monitoring the activity of a signaling pathway up to the point at which the overexpressed proteins are located [136].

Fluorescence and luminescence-based assays allow for the measurement of signaling events in live cells in real-time without the need for cell lysis and can detect small changes in signal modulation. Some assays, such as the sensors of the GCaMP8 family, can measure cellular events at the millisecond scale [66]. In addition, non-complementation assays do not depend on the spatial restrictions of protein fragment complementation.

Enzyme-based reporter systems, such as BiLC, which uses a luciferase reporter, usually have a high dynamic range that is useful for measuring small changes in signaling activity. Also, protease complementation-based reporter systems, such as split TEV and full TEV techniques, have an additional amplification step through their proteolytic activity [57,137]. However, transcriptional readouts, as used by split TEV and full TEV techniques, are not able to detect cellular signals in real-time. In contrast, they permit the multiplexing of assays within the same cell and of various cells grown in the same well of a cell culture vessel, thereby enabling the profiling of multiple cellular events simultaneously [127,133].

In BRET and FRET assays, the targeted proteins can be fused to the reporter proteins (i.e., fluorescent proteins, *Renilla* luciferase) in two ways: firstly, the targeted protein can be fused to either one of the two reporter proteins. However, the fusion proteins may have a preferred orientation for an optimal signal-to-noise ratio. Secondly, the reporter protein can be fused to the N or C-terminal end of cytosolic proteins, such as β-arrestin, while reporter proteins must be fused to the cytosolic C-terminus of the GPCR. The same principle applies to complementation assays, such as BiFC, BiLC, and split TEV, where fragments of the reporter protein must be tested for optimal orientation. To further enhance the performance of complementation assays, reporter proteins and fragments of complementation assays can be separated from the targeted proteins by linker sequences that are flexible and commonly consist of glycine and serine residues [138].

## 6. Conclusions and Future Perspective

Various genetically encoded techniques are available to measure GPCR activity and their downstream signaling. These techniques capture signals at different stages of cellular signaling, from receptor activation to transcription, providing fine-tuned options for monitoring drug effects in living cells. Genetically encoded assays can use a wide variety of methods, such as FRET, BRET, or TEV protease-based methods and, when applied in singleplex format, use mostly fluorescence and luminescence as readouts. Multiplex GPCR assays extend the possibilities to the simultaneous monitoring of various GPCRs and signaling events throughout the cellular signaling cascades. This is achieved by compatible fluorescent or luminescent reporters, and, particularly, genetic barcodes in combination with NGS. The latter method enables the analysis of thousands of unique barcodes from one experiment and is therefore expected to have the highest potential for future multiplex assays and broad screening campaigns. The use of multiplex GPCR assays will continue to accelerate the deconvolution of the complex physiology and pharmacology that underlies ligand and drug actions mediated by GPCRs. Ultimately, the simultaneous measurement of GPCR activity, pathway promiscuity, and transcriptional regulation will help in identifying safer therapeutics.

## Figures and Tables

**Figure 1 ijms-25-05474-f001:**
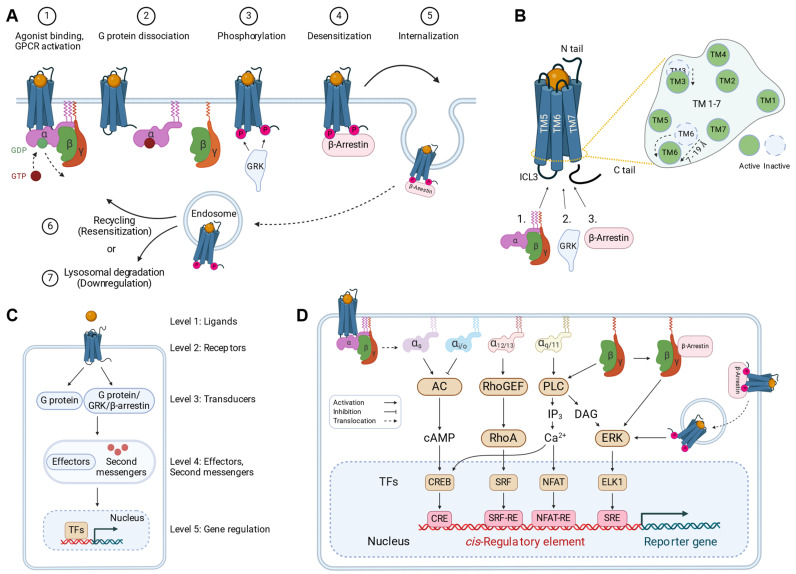
GPCR activation and downstream signaling can be monitored at various hierarchical levels of cellular signaling. (**A**) Canonical GPCR activation by agonists. (1) Agonist binding to GPCR/G protein complexes induces a conformational shift that leads to the release of GDP from Gα and the binding of GTP. (2) The GTP-bound active Gα and the Gβγ complex dissociate from the receptor and induce downstream signaling. (3) GPCR kinases (GRKs) then phosphorylate active GPCRs to allow arrestins to bind, resulting in GPCR desensitization. (4) Arrestins can also act as scaffolding proteins for GPCR internalization via clathrin/AP-2 or caveolin into early endosomes. (5) GPCRs can signal intracellularly from early endosomes, be recycled back to the plasma membrane (i.e., resensitization), (6) be recycled back to the plasma membrane (i.e., resensitization), or (7) be degraded in the lysosome (i.e., downregulation). (**B**) GPCR activation mechanism. GPCRs activate G proteins upon agonist binding, followed by phosphorylation via GPCR kinase (GRK), and subsequent β-arrestin binding. The intracellular loop 3 (ICL3) and the C-terminal tail (C-tail) are the primary regions for these three main transducers to bind and interact with. On the right: extracellular view at the intracellular surface of the GPCR with the movements of the transmembrane domain (TM) TM3 and TM6 during the activation event. TM6 and the connected ICL3 show a prominent outward movement. (**C**) Intracellular signaling can be monitored by assays at different levels. (**D**) Assay targets. GPCR-mediated signaling that can be monitored in cell-based assays. AC, adenylate cyclase; cAMP, cyclic adenosine monophosphate; CRE, cAMP-responsive element; CREB, cAMP-responsive element-binding protein; DAG, diacylglycerol; ELK1, erythroblast transformation specific-like protein-1; ERK, extracellular signal-regulated kinase; GDP, guanosine diphosphate; GTP, guanosine triphosphate; IP3, inositol trisphosphate; NFAT, nuclear factor of activated T cells; P, phosphorylation modification; PLC, phospholipase C; PM, plasma membrane; RE, response element; RhoA, Ras homolog family member A; RhoGEF, Rho guanine nucleotide exchange factor; SRE, serum response element; SRF, serum response factor; TF, transcription factor.

**Figure 3 ijms-25-05474-f003:**
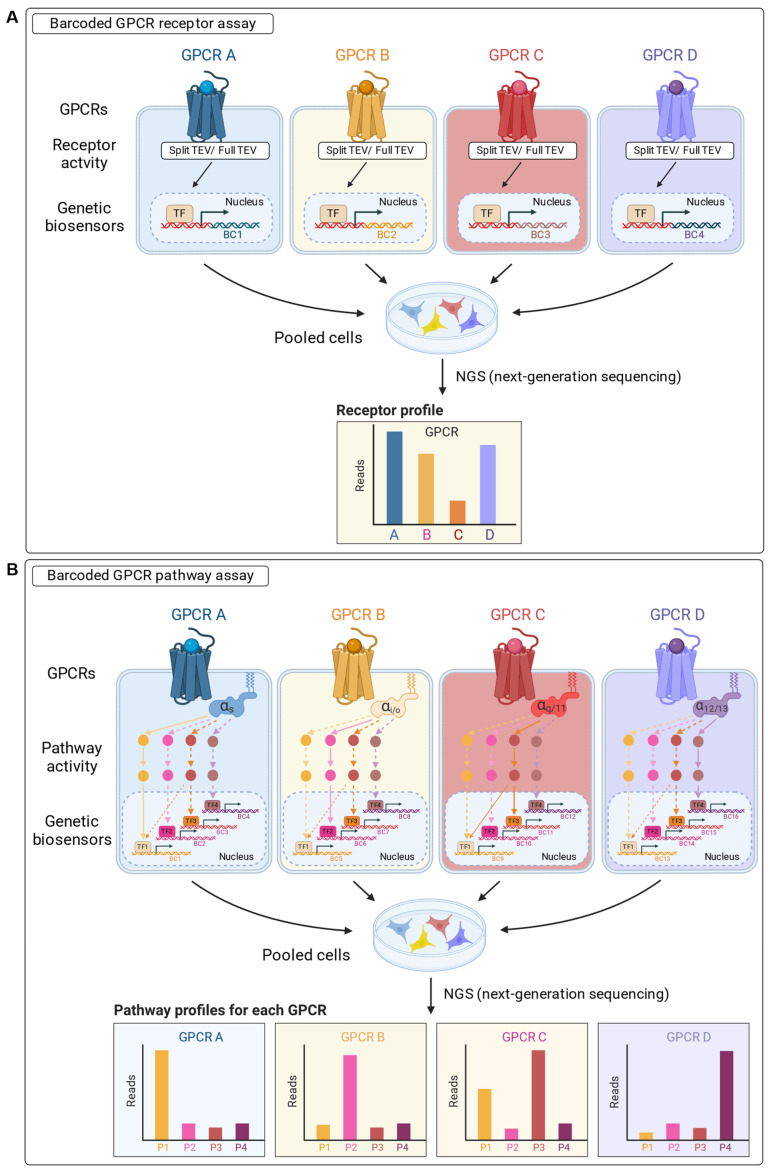
Multiplexed GPCR assays with barcoded readouts. (**A**) Barcoded GPCR receptor assay. Split TEV or full TEV β-arrestin-2 recruitment assays are used to monitor the activity of GPCRs [130,131]. The assay includes multiple GPCRs, each co-expressed with a transcriptional reporter containing a unique molecular barcode (BC) sequence. This links each GPCR to a response and allows for the simultaneous treatment of multiple GPCRs in a single well with pooled batches of cells. Barcodes are analyzed using next-generation sequencing (NGS) to generate an activity profile for multiple GPCRs per treatment. (**B**) Barcoded GPCR pathway assay. Multiple pathway activities are monitored for each GPCR using responsive *cis*-regulatory elements that capture the upstream activities of signaling pathways, such as the cAMP-responsive element (CRE), serum response element (SRE), or the calcium sensor nuclear factor of activated T cells response element (NFAT-RE). As for the barcoded GPCR receptor assay in (**A**), multiple GPCRs can be assessed simultaneously, resulting in a pathway profile for each GPCR per treatment. Note that both types of barcoded GPCR assays can also be combined [133].

**Table 1 ijms-25-05474-t001:** Genetically encoded singleplex assays for monitoring GPCR activity and signaling.

Hierarchical Level	Target Mechanism	Assay Technique	Readout	References
Receptor	GPCR conformational change	BRET	Fluorescence, Luminescence	[46]
	GPCR conformational change	Permutated GFPs	Fluorescence	[47,48]
	GPCR dimerization	FRET	Fluorescence	[49]
	GPCR dimerization	BRET	Fluorescence, Luminescence	[50]
	GPCR endocytosis	FRET	Fluorescence	[17]
	GPCR endocytosis	BRET	Fluorescence, Luminescence	[51]
Transducer	G protein recruitment	FRET	Fluorescence	[52]
	G protein recruitment	BRET	Fluorescence, Luminescence	[53,54]
	β-arrestin recruitment	FRET	Fluorescence	[55]
	β-arrestin recruitment	BRET	Fluorescence, Luminescence	[46,55]
	β-arrestin-2 recruitment	Full TEV assay (Tango)	Luminescence	[56,57]
	β-arrestin-2 recruitment	Split TEV assay	Luminescence	[58]
	Gα recruitment	TGF-α shedding assay	Absorbance	[59,60]
Second messenger	cAMP/Epac	FRET	Fluorescence	[61,62]
	cAMP	GloSensor	Luminescence	[63,64]
	Calcium/Calcium flux	GCaMP	Fluorescence	[65,66]
Effector	RhoA	FRET	Fluorescence	[67]
	RhoA	Relocation	Fluorescence	[68]
Transcription	Transcription factors (CREB, NFAT, ELK1, SRF)	Reporter gene activation (Pathway assay)	Luminescence	[45]

Abbreviations: Epac, exchange protein directly activated by cAMP; GFP, green fluorescent protein; TEV, tobacco etch virus protease.

## Data Availability

Not applicable.

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
