# Peer review of "Exploiting Cell-Based Assays to Accelerate Drug Development for G Protein-Coupled Receptors"

_ijms, 2024, doi:10.3390/ijms25105474_

Round 1

Reviewer 1 Report

Comments and Suggestions for Authors

General Comments:

In this review article, the authors describe the technology and use of various "engineered-cell" assays for measuring GPCR-mediated responses. The article is well organized and very clearly written, with excellent explanations of each technique.

Major Comments:

1. Some aspects are presented a bit simplistically. For example, Gi/o does not necessarily signal by reducing cAMP, but also through Gbeta/gamma-mediated effects on other effectors.

2. All of the assays described in this review utilize "engineered" cellular models. Such systems can be quite expensive and are not always practical for small-scale studies. Also, they do not necessarily reproduce the responses seen when receptors are not over-expressed. The expression levels of G-proteins and other effectors can be limiting for the response; this aspect is not discussed. The profile of responses seen in these systems can be extremely useful, but is not necessarily representative of what will be seen in more physiological models and cell types. It is suggested that the authors provide more perspective on the scope of the review at the outset, and also include possible pitfalls in the utilization of these high-throughput assay systems. In other words, it would be helpful if the advantages and disadvantages of these approaches were presented.

3. Figure 2 is beautifully laid out, but the authors are trying to show too much in one figure. The panels are too small as presented, and the overall figure is overwhelming to the reader. The legend takes up nearly an entire page. It may be possible to split this figure up into multiple figures.

Reviewer 2 Report

Comments and Suggestions for Authors

The review manuscript authored by Yuxin Wu et al. provides a comprehensive overview of genetically encoded cell-based assays for measuring GPCR activation and downstream signaling at different hierarchical levels. The manuscript is well organized, informative and complemented by clear schematic illustrations.

Minor points:

Receptor-Heteromer Investigation Technology (Receptor-HIT) could be added as a potential BRET-based tool to detect GPCR heteromers.

Minor deficiencies include some typos (e.g., missing spaces after "orientation" in line 85 and "ligands" in line 259). There are also instances where the text appears to be unnecessarily bolded (e.g. in lines 377-379 and 509-512).

Round 2

Reviewer 1 Report

Comments and Suggestions for Authors

The authors have responded appropriately to the comments made in the first round of reviews. The manuscript is a useful addition to the literature, and will particularly be appreciated by graduate students and postdocs who are approaching these assays for the first time.